# Maintaining essential healthcare services in Addis Ababa during COVID-19: A qualitative study

Esete Habtemariam Fenta[1], Berhan Tassew[1]\*, Admas Abera[2], Firmaye Bogale Wolde[3], Meseret Legesse[1], Justin Pulford[4], Siobhan Mor[5,6], Mirgissa Kaba[1]

**1** School of Public Health, Addis Ababa University, Addis Ababa, Ethiopia, **2** Epidemiology and Biostatistics Unit, School of Public Health, Haramaya University, Harar, Ethiopia, **3** Knowledge Translation Directorate, Ethiopian Public Health Institute, Addis Ababa, Ethiopia, **4** Department of International Public Health, Liverpool School of Tropical Medicine, Liverpool, United Kingdom, **5** Institute of Infection, Veterinary and Ecological Sciences, University of Liverpool, Liverpool, United Kingdom, **6** International Livestock Research Institute, Addis Ababa, Ethiopia

\* tassewberhan@gmail.com

**Data Availability Statement:** All relevant data are within the manuscript and its Supporting Information files.

## Abstract

### Background

Worldwide, health systems have been challenged by the overwhelming demands of the COVID-19 pandemic. In Ethiopia, maintaining essential health services during the COVID-19 pandemic is critical to preventing severe outcomes and protecting the gains made over the past years in the health sector. This project aims to explore the health system's response to maintaining essential healthcare services in Addis Ababa, Ethiopia.

### Methods

A total of 60 key informant interviews were conducted by purposively selecting key stakeholders from Federal Ministry of Health, Addis Ababa Regional Health Bureau, Sub-city Health Offices, and frontline healthcare providers. Interviews were transcribed verbatim and coded using Open Code. Thematic analysis was employed to analyze the data.

### Result

COVID-19 affected the delivery of essential health services in several ways, namely: decline in health service utilization, fear of infection among healthcare providers, stigma towards healthcare providers, and perceived decrease in quality-of-service provision. However, the health system actors made efforts to sustain services while responding to the pandemic by enacting changes in the service delivery modality. The most significant service delivery changes included repurposing health centers and prolonged prescriptions (multi-month medication dispensing). The primary challenges encountered were burnout of the health workforce and a shortage of personal protective equipment.

**Funding:** This research was supported by the Global Challenges Research Fund (GCRF) One Health Regional Network for the Horn of Africa (HORN) Project, from UK Research and Innovation (UKRI) and Biotechnology and Biological Sciences Research Council (BBSRC) (project number BB/P027954/1). The funders had no role in study design, data collection and analysis, decision to publish, or preparation of the manuscript. There was no additional external funding received for this study.

**Competing interests:** The authors have declared that no competing interests exist.

**Abbreviations:** ANC, Antenatal Care; ART, Antiretroviral Therapy; CAG, Community Antiretroviral Therapy Group; DOT, Directly Observed Therapy; FMOH, Federal Ministry of Health; HC, Health Center; HEW, Health Extension Worker; HIV, Human Immunodeficiency Virus; IPC, Infection Prevention Control; NCD, Non-Communicable Diseases; NGO, Non-Governmental Organization; PHCU, Primary Health Care Unit; PPE, Personal Protective Equipment; TB, Tuberculosis.

## Conclusion

COVID-19 has affected the delivery of essential health services in multifaceted ways. System actors have accordingly made efforts to sustain services while responding to the pandemic.

## Background

After the global COVID-19 outbreak in December 2019, Ethiopia reported its first positive case on March 13, 2020. As of March 28, 2021, a total of 200,563 confirmed COVID-19 cases and 2801 deaths were recorded in the country [1]. Globally, health systems have been challenged by the overwhelming demands of the COVID-19 pandemic [2–9]. Responding to the healthcare burden imposed by the pandemic required intense resources, as this had to be coupled with ensuring continued access to essential healthcare services [10]. Countries struggled to allocate resources for these needs because health facilities operated with less capacity due to the distribution of resources, including healthcare providers for COVID-19 prevention, detection, and case management [11, 12]. Moreover, infection among healthcare providers resulted in missed working days and a backlog of care [13, 14].

Worldwide, almost every country (90%) experienced disruption to its health service, where low-and middle-income countries reported greater difficulties [15, 16]. The most frequently (more than 50%) disrupted essential health services were routine immunization services, outreach services and facility-based services, noncommunicable disease diagnosis and treatment, family planning and contraception, treatment for mental health disorders, antenatal care, and cancer diagnosis and treatment. The disruptions mainly occurred more at the earlier stage of the pandemic and reduced with time [17, 18]. In addition to mortality and morbidity directly attributed to COVID-19, the pandemic poses a significant risk of indirect morbidity and mortality from other preventable and treatable diseases if essential health services are disrupted, as might occur when available healthcare resources are diverted to fight the pandemic [19].

The World Health Organization recommended ensuring the continuation of essential services, including vaccination, chronic disease follow-up, and maternal and child health care, by taking the local context and extent of the outbreak into consideration [20]. Maintaining essential health services (i.e., Providing an uninterrupted essential healthcare service to communities) during the COVID-19 pandemic is critical to prevent severe outcomes and protect the gains made over the past years in the health sector. In recognition of the burden of the pandemic on the health sector, the Ethiopian Federal Ministry of Health (FMOH) developed national guidelines for managing COVID-19 in April 2020. The guideline set standards for surveillance, tracing protocols, COVID-19 treatment centers, as well as health facilities preparedness, community engagement, and maintaining essential services during the pandemic [21]. The Ethiopian essential health service package includes reproductive, maternal, neonatal, child, and adolescent health services; major communicable diseases; non-communicable diseases; surgical care; and emergency and critical care [22].

The pandemic has exposed the limitations of many health systems, including those previously classified as high-performing and resilient [23]. While it has been demonstrated that COVID has a negative impact on health systems, less is known about its effect on health facilities serving large populations in an urban setting like Addis Ababa and their provision of essential healthcare services. A comprehensive analysis of the health sector's resilience during the pandemic can therefore pinpoint important lessons and help strengthen countries'

preparedness, response, and approach to future health challenges [20]. The findings of this study can also provide important context for decision/policy making to improve planning and coping mechanisms of similar public health emergencies and revitalize the essential healthcare services post-pandemic. Hence, this study was conducted with the aim of exploring the mitigation strategies (actions taken) by the health sector to ensure sustained (uninterrupted) provision essential healthcare services during COVID-19 pandemic in Addis Ababa, Ethiopia.

## Methods and materials

### Study design and setting

This study employed an exploratory qualitative research design using open-ended interview question and a purposive sampling approach. This design uses participants' narratives to best articulate and understand the phenomenon of interest, especially when the issue is poorly understood. This study was conducted in the capital of Ethiopia, Addis Ababa, which has a population of 8,938,683 million (30% of the country's urban population). At the time of the research, the city had the largest number of reported COVID-19 cases in Ethiopia.

Ethiopia has a three-tier healthcare delivery system: level one is a Woreda/District health system comprised of a primary hospital (to cover 60,000–100,000 people), health centers (1/15,000–25,000 population) and their satellite Health Posts (1/3,000–5,000 population) connected to each other by a referral system. The primary hospital, health center, and health posts form a Primary Health Care Unit (PHCU). Level two is a General Hospital covering a population of 1–1.5 million people, and level three is a Specialized Hospital covering a population of 3.5–5 million people [24].

### Study approach and period

We conducted a qualitative study to explore the health system response to sustain the provision of essential healthcare services during the pandemic. This study was conducted from February to March 2021.

### Sample size and study participants

Participants were purposively selected based on the possibility of having a comprehensive information about the effect of COVID-19 and mitigation strategies and/ or the health system response to sustain the provision of essential healthcare service in their respective departments. Accordingly, study participants were based in Addis Ababa city where we purposively selected participants from the FMOH, Addis Ababa regional health bureau, and Sub-city health office who were assigned to the COVID-19 response team. In addition, we interviewed the medical director/focal person of the rapid response team and service providers at health centers from different contact points that provide essential health services. We determined the sample size based on the saturation of information. We included 60 key informants in this study and there were no non-participants in this study.

### Data collection tool and procedure

Interview guides were developed with open-ended questions to ensure high degree of flexibility. Interview guides covered topics such as the effect of COVID-19 on essential healthcare service provision and utilization, activities undertaken to sustain the provision of essential healthcare service, infection prevention and control (IPC) measures taken to ensure safe delivery of essential healthcare service, and challenges faced by the health system while providing essential healthcare service during the pandemic. We pilot-tested interview guides and made

the necessary refinements to make sure the guides could elicit the information we intended to capture. The pilot was conducted among healthcare providers that were not included in this study. Based on the pilot interviews, we paraphrased some of the questions and added probing questions to get detailed information.

Face-to-face interviews were arranged for a time and location that was deemed most favorable and comfortable for the participants. Interviews were conducted in a quite office and privacy was ensured to provide participants the freedom to express their opinions by conducting the interview in the presence of the interviewer and the respondent only. All interviews were audio taped and took about an hour on average. Five authors (4 female (EHF, BT, FBW, ML) and one male (AA) conducted the interviews. All interviewers are MPH holders, trained in qualitative methods, and have prior experience in qualitative data collection and analysis. Among the interviewers, three (EHF, BT, AA) were lecturers, one (FBW) was a director of knowledge translation directorate, and one (ML) was a project coordinator at the time of data collection.

During data collection, respondent validation was done by restating and paraphrasing information for respondents to determine the accuracy. Additionally, investigator triangulation was applied by using multiple investigators to study the same phenomena improving the credibility of findings of the study.

## Data analysis

Thematic data analysis was used to describe and compare general statements as relationships and themes present on the data [25]. The interviews were first transcribed verbatim into Amharic then translated into English. Inductive and deductive processes were used to identify common themes. Data analysis started with reading and re-reading each transcript and listening to the audio recordings. Once this was done, a codebook was developed by a study team of five researchers/coders. The study team double-coded 4 randomly selected transcripts and reviewed them together. If coder disagreement revealed ambiguity in codes, code definitions were updated, and examples were added. Double coding proceeded until no new disagreements were identified and inter-coder agreement was reached. After this, the transcripts were single-coded using open code (V 4.02). Finally, decision was made on the themes that gave possible explanations and made meaningful contribution to answering the research question.

## Rigor

The trustworthiness of the data was ensured using different approaches, namely, credibility, dependability, confirmability, and transferability. To ensure credibility, data was collected from different health system levels, starting from the Ministry of Health down to health centers, and included individuals from different contact points that provide essential health services. The last author, who has rich experience in qualitative research, oversaw the codebook development, data coding, and category development. Interviews were conducted by five primary and co-primary investigators. The interviewers are trained in qualitative methods and have prior experience conducting interviews and analyzing qualitative data. During data collection member checking was done by restating, paraphrasing, and summarizing information for respondents to determine the accuracy of data. To ensure the dependability of the findings, more than one individual was involved during both data collection and coding, and debriefing was conducted daily. Furthermore, the transcribed data were systematically reviewed against the audio files. Transferability of the data was maintained by a rich description of the method section of the study. The study finding confirmability was guaranteed with the research

supported by the data collected. Data were collected from various contact points to ensure all views were represented. Interviews were continued until data saturation was reached.

### Ethical consideration

Ethical clearance was obtained from the Research Ethical Committee of the School of Public Health, Addis Ababa University (reference number 070/20SPH) and the Ethical committee of the University of Liverpool (reference number 8049). In addition, permission was obtained from relevant federal and regional health offices. Written informed consent was obtained from the participants after the necessary explanation about the purpose, procedures, benefits, and risks of the study had been made. The informed consent also included permission to audio record the interviews and use anonymized quotes. The respondent's right to refuse few or all the questions was respected at all times. In addition, participants' privacy and confidentiality of the information obtained were always maintained.

## Result

Among the respondents, 53.34% (32/60) were female and 46.66% (28/60) were male. The age of participants was between 25 & 45, with a mean age of 31.5 (± 4.6 SD). Professionally, 51.6% (31) were public health professionals, 23.34% (14) were nurses, 16.67% (10) were midwives and 8.33% (5) were medical doctors.

The findings were structured into sections according to the two themes and ten sub-themes. The main themes and sub-themes were, effect of COVID-19 (effect on service utilization, effect on healthcare providers and effect on service provision) and adaptation strategies (designating facilities as COVID-19 isolation and treatment centers, prolonged prescription, community based health service, prioritization, ensuring availability of resources, support, infection prevention control (IPC) measures.

### Theme 1: Effect of COVID-19

**Effect on service utilization.** Participants explained COVID-19 resulted in an overall reduction in patient flow, loss-to-follow-up and defaulting on treatment. This was especially true in the first few months after COVID-19 was first reported in Ethiopia.

"*During COVID-19, clients did not want to come to HC at all because they thought they would get COVID in addition to their health problem. The patient flow was almost close to zero at one time. But once the effect of COVID decreased the panic among the public also started decreasing and patients started coming back to health center (HC). That's when we saw patients who have stopped their medication.*" (Male, 34, Medical doctor)

Our finding indicated that clients in Antiretroviral Treatment (ART) clinics were missing their appointments causing clients to suffer from opportunistic infections.

" *ART clients did not come here as they used to since they are more susceptible to COVID-19 due to their immune status. There were also some clients who were lost-to-follow-up due to fear of contracting the infection by coming here.*" (Male, 35, Public Health)

Similarly, the number of patients with Non-Communicable Disease (NCD) who came for follow-up significantly reduced due to fear of contracting COVID-19 at health facilities. Some patients were known to have discontinued their medication while others started buying

medication from private pharmacies without follow-up. This caused patients with NCD to suffer from complications. A participant mentioned the situation as follows:

"...we saw some patients who have stopped taking medication. The number of people with complications like diabetic ketoacidosis (DKA) and hypertensive emergency has also increased because of not having follow-up." (Male, 34, Medical doctor)

Our findings also indicated that patients with Tuberculosis (TB) refused to come for follow-up and even those coming to the health facilities did not want to stay long for physical examination and laboratory investigation. In addition, people who had cough were hiding their symptoms and using home remedies hoping they would get better from fear of being diagnosed with COVID-19 and being quarantined. This led to late detection and low diagnosis of TB during the earlier phase of COVID-19 in the country.

Participants indicated that the number of visits for delivery, family planning and first antenatal care visit had shown a decrement. Participants mentioned that the number of unwanted pregnancies and abortion increased. Similarly, pregnant mothers delayed or missed their appointments to avoid exposure. In addition, participants indicated that there were home deliveries.

"Yes, there were home deliveries in our Woreda. There were some mothers who had follow-up for seven months or so and delivered at home. The number of home delivery even exceeded last year's home delivery. (Male, 37, Midwife)

Participants mentioned that families were not willing to bring their children for immunization due to fear of infection at health facilities. In addition, some were reluctant to get their children vaccinated during campaigns for childhood diseases thinking it would be a trial for a COVID-19 vaccine. One participant explained this as follows:

"At the beginning of the pandemic there were vaccination campaigns for measles and polio, and we have seen fear among the community. They did not believe we were providing polio, or measles vaccines. Thus, they refused to get their children vaccinated and said they do not want to vaccinate their children because of fear." (Female, 28, Nurse)

Our findings also indicated that health service utilization by people with conditions that did not require regular follow-up also decreased. People used to visit health facilities for minor complaints such as headache. However, most people preferred to use home remedies instead of visiting a health facility during COVID-19.

"After COVID-19 was reported the maximum number of patients we see per day was down to five. When we ask people why they are not coming, they tell as they are taking traditional medicines like garlic and garden cress." (Female, 28, Public Health)

**Effect on healthcare providers.** Participants indicated that fear among health professionals was common due to the limited supply of PPE and small examination rooms to ensure social distancing. In addition, health professionals had limited information about prevention and management of COVID-19, and they didn't have formal training. Rather, they were exposed to national and international media which explained the severity of the disease. This created fear, frustration, and anxiety among the providers. One participant indicated the situation as follows:

"*We were equally scared of the community as they were scared of us. The community feared us, and we feared the community.*" *(Male, 29, Public Health)*

In addition, COVID-19 created extra workload on the health professionals since they had additional COVID-19 response activities including screening at the gate and working in isolation rooms in addition to the routine services. Health providers were also required to serve clients beyond their usual catchment area when adjunct health facilities were converted to COVID-19 treatment/isolation centers. This resulted in high client flow in adjunct health facilities which resulted in increased workload.

Our findings indicated that health providers were discriminated by the community as they were considered a source of infection. A participant explained the situation as follows:

"*The community wasn't very accepting of us, they used to mistreat the health professionals. Some even referred health professionals as "The COVID-19s". The community used to say "The COVID-19s are here" when we go for home visits. We had a hard time.*" (Male, 38, MPH)

**Effect on health services provision.** Our findings indicate that COVID-19 affected the quality of health services. Overall, COVID-19 affected patient-physician communication since some providers preferred to give services quickly to reduce their exposure. As a result, physical examination, and counseling services were compromised. One participant explained this as follows:

"*COVID-19 affected the service. I say this because we were limiting our contact hour with the clients. Immunization is a service where you provide vaccines, schedule the next appointment, monitor growth, provide counselling about side effects, counsel about infection prevention, and so on. This means it needs a lot of time to cover everything, but we were limiting our contact hours to prevent the infection, so all these services were compromised.*" *(Female, 32, Nurse)*

Some participants reported that the quality of service given for ART patients has been compromised. This is because newly diagnosed Human Immunodeficiency Virus (HIV) patients used to be given treatment after repeated counseling with service providers and case manager to ensure their readiness and ability to sustain lifelong treatment. However, to minimize contact, new patients were prescribed medication on the same day they were diagnosed without such careful assessment of their readiness.

"*I have been working on ART unit for long and I can tell you that the quality of ART service has been compromised. We are giving ART medication for new HIV patients without checking whether they have accepted it or not. There are patients who need time to decide what they want to do, and previously we used to talk to them, and the case manager used to have repeated discussion with them before we put them on medication. But now we put them on medication on the same day they are diagnosed.*" *(Female, 35, Public health)*

There was also a gap in monitoring viral load because of delay in laboratory test. Thus, patients stayed on medication that had to be changed since treatment regimen is based on viral load.

"*The other challenge that I want to tell you is related with viral load testing. We don't know where our samples are. We get viral load test done at Ethiopia Public Health Institute but during the pandemic more focus was given for COVID sample, so we were getting incomplete results for the samples that we have sent. We even collected sample from our clients for a second round to get the viral load result. There was more than 6 months delay to get the result for viral load. This was the other main challenge that COVID posed on ART service.*" *(Female, 35, Public Health)*

Different investigations including organ function testing was also compromised because of a shortage of reagents. As a result, providers relied on signs and symptoms reported by the patients rather than laboratory tests to follow their patients.

### Theme 2: Adaptation strategies to sustain the provision of essential healthcare services

**Designating facilities as COVID-19 isolation and treatment centers.** As part of the pandemic response, one or two health centers from each sub-city were designated as COVID-19 isolation and/or treatment centers. This meant that non-COVID-19 cases and regular clients must go to alternate health centers. The selection of the health center as isolation and/or treatment center was based on experience of the health centers to manage outbreaks that have happened before. In addition, they selected health centers where the patient flow is low.

Challenge with this adaptation strategy was overcrowding of adjunct health facilities and burn out of healthcare providers as they were serving people from outside of their catchment areas where the health centers have been assigned as COVID-19 center.

**Prolonged prescription (multi-month medication dispensing).** Participants explained that prolonged prescription of medication was the primary alternative mode of care that health centers used to sustain provision of different services. Health facilities started implementing prolonged prescriptions once direction was given by Ministry of Health, which was around 1 to 2 months after COVID-19 was first reported in Ethiopia. Prolonged prescription was implemented in all heath centers primarily for clients with follow-up such as RVI, TB and NCD patients. The period of prescription varied across different services. For instance, new RVI patients were given 3 months of supply during COVID. This, however, was not the practice before COVID-19 where new ART patients were given medication for two weeks in two rounds, then monthly medication followed by three months medication and finally six months medication by appointment spacing model designed for stable patients regularly checking their adherence and status.

"*We were given direction to use prolonged prescription. Therefore, what we did was, we took out all the patients' card and we started calling our clients so that they can come and take their medication. We were giving them supply that would be enough for 3 months if they are new and 6 months for the rest of our clients.*" *(Male, 31, Public health)*

Regarding TB services, health facilities previously followed Directly Observed Therapy (DOTs) during the first two months of intensive treatment, during which time clients were expected to attend the health center daily. However, during the pandemic this modality was changed and clients in the intensive phase were given either a week or two weeks of supply depending on their condition. Likewise, during the second (continuation) phase of TB management clients normally come weekly for treatment. During the pandemic, this was changed so that clients were getting a monthly supply during this phase.

Similarly, NCD clients were given 3–6 months of supply to minimize frequency of contact. With this modality, health care providers were monitoring adherence of clients to their medication through phone calls, self-report during follow-up appointments and through viral load testing for ART clients.

The main reported challenges related with prolonged prescription include poor monitoring of adherence and poor adherence among clients resulting in treatment failure and complications of different cases. Prolonged prescription has also resulted in a shortage of medication at the health center. Though healthcare providers prescribed three months' supply for NCD clients, there was a shortage of supply at the health centers. As a result, clients were only able to purchase a one-month supply from other governmental pharmacies.

**Community based health service.**   The other mitigation strategy implemented by the health centers as explained by the participants was community-based provision of different services through primary health care team. This team is comprised of different service providers from different units including Health Extension Workers (HEW). Though this system was in place before COVID-19, it was strengthened during the pandemic.

Some of the services provided by the team as explained by the participants included screening for NCD and TB cases, tracing loss to follow-up, provision of health education, distribution of Iron Folic Acid supplementation for pregnant women and vitamin A supplementation for children. This was captured as follows:

"*There is what we call primary health care team, and this team goes out to the community 4 days in a week. During community visit, health care provider in the team in collaboration with the HEWs create awareness about COVID-19 and other medical conditions. There is also provision of medication and family planning service for those who can't come to health facility.*" (Male, 31, Midwife)

Participants explained community-based distribution of medication was used for ART clients through a system called "Community ART Group (CAG)". This system was in place before COVID-19, but it was strengthened during the pandemic through collaboration with non-governmental organizations. With this system, medications were distributed at community level either at school or church. This approach is captured as follows:

"*Using the community ART group, we (providers) arrange the community in a group of 5 or 6 then we go to the community and distribute the medication for the group. We were working in collaboration with Non-governmental Organization (NGO), and they allocated a car for us to deliver the service for the community.*" (Female, 35, Public Health)

### Prioritization

Participants explained that health facilities prioritized certain services with the aim of freeing up the space and resource for the provision of other essential healthcare services. There were facilities that suspended services such as, medical service for a driving license, cervical cancer screening, minor surgeries such as circumcision. However, this only lasted for the first two to three months of the pandemic, and they resumed providing the service by taking the necessary precautions. One participant explained this as:

". . ..*In addition, we suspended services like circumcision service and cervical screening service since they were not urgent. Service for general medical checkup was also suspended at first but*

*then we started providing it because we believed we can sustain the provision by taking the necessary precaution. Now every service is being given." (Male, 29, Public Health)*

Participants also explained that prioritization of clients was in place. All facilities prioritized clients who were believed to have high risk of getting seriously ill from COVID-19. These included patients with underlying medical conditions such as ART and NCD patients, pregnant women, and children. These clients were given priority to minimize waiting time to access services, thereby reducing their duration of stay at the HC. This is captured in the following quote:

*"The medical card of ART patients is kept in the case manager office, so they don't have to wait in line to get their card. Thus, clients are able go straight to the case manager office and get the service in ART clinic which is located right next to the case manager office. This means they immediately get the service and leave." (Female, 35, Public Health)*

The other strategy implemented by the health facilities to prioritize clients coming for ART services was through changes in work-hour arrangements. Providers started working early in the morning, during lunchtime and during Saturdays to provide ART service and to minimize crowding.

### Ensuring availability of resources

Health centers took different measures to ensure adequate workforce and supply to sustain essential healthcare services. In terms of workforce, no service provider was allowed to take annual leave except for those who were high risk. These included service providers who were pregnant, elderly or who had comorbidities. In addition, providers on study leave were asked to return to work.

In terms of supplies, health centers used to borrow supplies from one another when they faced shortages. This includes supplies needed for essential services as well as personal protective equipment (PPE). Health facilities also reshuffled budgets such as shifting budget allocated for training to purchase PPE.

### Support

According to our findings, different governmental and non-governmental organizations provided support for health facilities. Non-governmental organizations supported facilities by providing vehicles for CAG services, and mobile airtime cards used to trace clients and remind upcoming appointments. In addition, they provided logistic support such as masks, sanitizers, soap, and tents.

Federal and governmental offices also provided support for health centers by providing o masks and sanitizer. However, participants explained that the support from sub-city in terms of supportive supervision to ensure sustained provision of essential healthcare services reduced significantly.

*"Higher offices should have their own contribution to ensuring that services are being provided properly. There should have been supervision to ensure that services are being provided properly and that the community is not inconvenienced but no one came from higher offices to supervise because of fear of contracting an infection when they come to health center. People might be scared, and it is understandable, however, they should have monitored whether essential services were being provided properly or not." (Female, 40, MPH)*

*"We (personnel at sub-city) did not do that much monitoring and supervision. The professionals here at the sub-city did not want to go to health centers to monitor or supervise since there was frustration that resulted from fear of being infected from going to health facilities. So, one of the drawbacks of COVID-19 was the visible collapse of service deliveries, because of the poor support and supervision of health services." (Male, 36, MPH)*

### Infection Prevention Control (IPC) measures

Initiative of renovating facilities for infection prevention including securing water supplies was done by sub cities by following strict application of guidelines.

Regarding providers' IPC implementation practice, most healthcare providers explained that there was a stricter adherence to infection prevention activities like wearing masks and washing or sanitizing hands. However, the use of masks and sanitizers has loosened over time.

*"We use sanitizer and mask, but we hug and kiss. We also used to eat lunch separately by keeping distance, but now we are eating in one dish, which was the custom before COVID-19; everything is getting back to normal." (Female, 35, Public Health)*

As for IPC measures taken for and by patients, initially, temperature screening at the health facility entrance, isolation of clients with symptoms of COVID-19, social distancing in waiting areas, mandatory mask use and hand washing was enforced. There was also repurposing of rooms for better ventilation if the previously used rooms were too small. Although the IPC measures were strictly implemented during the first few months, practices have become lapsed as indicated by our participants.

Participants explained the different challenges they have faced related with IPC measures. The challenges cited include, lack of well-equipped isolation room, shortage of PPE including mask, sanitizer, glove, and water especially during the first 2–3 months resulting in frustration among service providers.

*"We had shortage of PPE and the budget that was allocated for health centers was very low this year. It was even lower than last year's budget, so we were challenged to equip the health center with the necessary logistics. In addition, the price of materials has gone up this year. A single surgical mask that used to cost around 3 birr now costs around 9 or 10 birr. So, we struggled to purchase a mask. We used to give providers one mask for a weak and half liter sanitizer for a month. This has made the providers feel uncomfortable." (Male, 37, MPH)*

Other challenges related to IPC measures were a shortage of isolation rooms, poor infrastructure of health facilities to ensure the safe delivery of different services, and negligence from the community/clients towards prevention strategies. There were also challenges related to sustaining the implementation of different IPC measures, including screening at the gate, ensuring clients wash their hands, and ensuring social distancing at the health centers.

Poor preparedness was one of the reasons identified by participants that has resulted in the shortage of supply.

*"We could have been more prepared. There was negligence starting from the higher health office. Otherwise, we would not have a shortage of logistics. We would not have such a shortage if we had thought about this earlier. If you go to the pharmaceutical supply agency, they will tell you that they don't have mask even though they have, and this shows you that they were not ready and they didn't have adequate stock." (Male, 29, Public Health)*

## Discussion

This study explored the health system response to maintain the provision of essential health-care services in Addis Ababa, Ethiopia during the COVID-19 pandemic. With this study, we identified the effect of COVID-19 on service utilization and provision. We also identified key adaptation strategies implemented by the health sector to sustain the provision of essential healthcare services and the challenges encountered. Our study findings highlighted that COVID-19 affected the delivery of essential health services in various ways during the early stages of the pandemic. The pertinent effects were perceived decline in health facility utilization, fear of infection among healthcare providers, stigma towards healthcare providers, and a general decrease in the quality of services. Overall, health system actors made efforts to sustain services while responding to the pandemic through providing supplies, and enacting changes in the service delivery modality. The most significant changes in the service delivery included multi-month medication dispensing and repurposing of health centers. The primary challenges encountered were burnout of the health workforce and a shortage of PPE.

COVID-19 has challenged health service delivery and resulted in a reduction of healthcare services utilization globally [3–10, 16, 23, 26]. Our findings also indicated a perceived decrease in health facility visits to utilize essential health care services. The primary reason for this was perceived fear of infection among the community. This is supported by similar studies in other countries that emphasize the effect of fear on service utilization [27–31]. Fear was greatest in the earlier phase of the pandemic, where less was understood about the pandemic. Fear among the community might have been exacerbated by the media's portrayal of the pandemic [32]. In particular, the public was consistently advised that people with comorbidity were at higher risk of contracting infection. Such messages might have inadvertently discouraged people with co-morbidities from visiting health facilities for essential healthcare services. In addition, rumors in the community about quarantine/isolation might have impacted clients' decision not to visit health facilities for essential health services.

Our finding indicated fear was common among healthcare providers as well. This is supported by other studies that indicated fear to be the most common psychological reaction among healthcare providers who continue to provide healthcare services during the COVID-19 pandemic [30, 33–35]. Different reasons contributed to healthcare provider's fear, one of the most common reasons was lack of personal protective equipment, especially at the beginning of the pandemic. Shortage of PPE is also reported by other similar studies as a major challenge contributing to healthcare provider's fear [32, 36, 37].

Other reasons for fear include fear of stigma and fear of carrying the virus back home and infecting loved ones [33, 38]. Such psychological distress coupled with high workload demand among healthcare providers could result in burnout and compromise the quality of service. Hence, it is important for governments and policymakers to consider and allocate funding to promote and provide psychosocial support for healthcare providers during a pandemic.

Health system actors made an effort to sustain the use of essential healthcare services by enacting changes in service delivery. The primary change was the use of multi-month dispensing of medication (prolonged prescription) for up to six months for people who may be at increased risk of COVID-19, such as people living with comorbidity. Even though this modality can minimize the risk of COVID-19 infection, there is a chance that some patients might suffer from complications because of limited follow-up of the diseases. In addition, there was no mechanism to monitor medication adherence other than self-report, where the validity can be affected by the recall period. Since medication adherence plays an important role in optimizing health outcomes, a mechanism to monitor adherence should be put in place instead of relying on self-report when using prolonged prescriptions [39].

Designating health centers as COVID-19 treatment/isolation was another response by the health system. In this study, healthcare providers explain that though this approach appears to have been effective in ensuring the sustained provision of services, it has increased their workload. This is because providers in non-COVID-19 treatment/isolation centers were serving clients outside their catchment population. In addition, they were providing other COVID-19 prevention activities such as screening, isolating, and engaging in community-based information dissemination, which resulted in burnout. Similarly, burnout among health workforce has been identified as a challenge in other studies [13, 34, 40, 41]. Burnout among health care providers has been associated with poor quality of care [42]. Thus, it is crucial for organizational leaders and policy makers to mitigate and reduce burnout among healthcare providers during pandemic.

Our current study has limitations that need to be acknowledged. This study was conducted only in Addis Ababa city; therefore, the findings may not apply to other regions. The other limitation is that it lacks the community/patient perspective. The strength of this study is it involved stakeholders starting from the ministry level down to the frontline health workers, which provides a comprehensive view of experience. In addition, we interviewed service providers from different contact points, which enabled us to capture the mitigation strategies at different contact points.

## Conclusion and recommendation

This study highlights the effect of COVID-19 on the health system and the health sector's action during the pandemic's initial phase. COVID-19 presented multifaceted effects, mainly fear among healthcare providers, perceived reduction in healthcare utilization, and decreased quality in health service provision. Key adaptation strategies implemented by the health sector to maintain essential healthcare service provision were multi-month dispensing of medications and repurposing of health centers. Primary challenges encountered include a shortage of PPE and burnout of healthcare providers, which could ultimately compromise the quality of health services. Therefore, the various impacts of the pandemic on the availability and utilization of health services can be inferred as a learning point for similar situations. Moreover, the health sector should consider fostering the different adaptation strategies identified in this study and integrating quality planning into future emergency preparedness planning.

## Supporting information

**S1 File. Key informant interview guide used in this study.**
(DOCX)

**S1 Dataset. The minimal data set 1.**
(OPCX)

**S2 Dataset. The minimal data set 2.**
(OPCX)

**S3 Dataset. The minimal data set 3.**
(OPCX)

**S4 Dataset. The minimal data set 4.**
(OPCX)

**S5 Dataset. The minimal data set 5.**
(OPCX)

**S6 Dataset. The minimal data set 6.**
(OPCX)

**S7 Dataset. The minimal data set 7.**
(OPCX)

**S8 Dataset. The minimal data set 8.**
(OPCX)

**S9 Dataset. The minimal data set 9.**
(OPCX)

**S10 Dataset. The minimal data set 10.**
(OPCX)

**S11 Dataset. The minimal data set 11.**
(OPCX)

**S12 Dataset. The minimal data set 12.**
(OPCX)

**S13 Dataset. The minimal data set 13.**
(OPCX)

**S14 Dataset. The minimal data set 14.**
(OPCX)

**S15 Dataset. The minimal data set 15.**
(OPCX)

**S16 Dataset. The minimal data set 16.**
(OPCX)

**S17 Dataset. The minimal data set 17.**
(OPCX)

**S18 Dataset. The minimal data set 18.**
(OPCX)

**S19 Dataset. The minimal data set 19.**
(OPCX)

**S20 Dataset. The minimal data set 20.**
(OPCX)

**S21 Dataset. The minimal data set 21.**
(OPCX)

## Acknowledgments

We would like to thank the HORN regional network for the Horn of Africa program for making this study possible by providing both financial and technical support and platform. We would like to extend our gratitude to our mentors for guiding the conduction of this study. We also want to thank the participants of this study without whom the study would not have been a reality.

## Author Contributions

**Conceptualization:** Esete Habtemariam Fenta, Berhan Tassew, Admas Abera, Firmaye Bogale Wolde, Meseret Legesse, Justin Pulford, Siobhan Mor, Mirgissa Kaba.

**Data curation:** Esete Habtemariam Fenta, Berhan Tassew, Admas Abera, Firmaye Bogale Wolde, Meseret Legesse.

**Formal analysis:** Esete Habtemariam Fenta, Berhan Tassew, Admas Abera, Firmaye Bogale Wolde, Meseret Legesse.

**Funding acquisition:** Siobhan Mor.

**Investigation:** Esete Habtemariam Fenta, Berhan Tassew, Admas Abera, Firmaye Bogale Wolde, Meseret Legesse.

**Methodology:** Esete Habtemariam Fenta, Berhan Tassew, Admas Abera, Firmaye Bogale Wolde, Meseret Legesse, Justin Pulford, Siobhan Mor, Mirgissa Kaba.

**Project administration:** Esete Habtemariam Fenta, Berhan Tassew, Admas Abera, Firmaye Bogale Wolde, Meseret Legesse.

**Resources:** Esete Habtemariam Fenta, Berhan Tassew, Admas Abera, Firmaye Bogale Wolde, Meseret Legesse.

**Software:** Esete Habtemariam Fenta, Berhan Tassew, Admas Abera, Firmaye Bogale Wolde, Meseret Legesse.

**Supervision:** Esete Habtemariam Fenta, Berhan Tassew, Admas Abera, Firmaye Bogale Wolde, Meseret Legesse, Justin Pulford, Mirgissa Kaba.

**Validation:** Esete Habtemariam Fenta, Berhan Tassew, Admas Abera, Firmaye Bogale Wolde, Meseret Legesse.

**Visualization:** Esete Habtemariam Fenta, Berhan Tassew, Admas Abera, Firmaye Bogale Wolde, Meseret Legesse.

**Writing – original draft:** Esete Habtemariam Fenta, Berhan Tassew, Admas Abera, Firmaye Bogale Wolde, Meseret Legesse.

**Writing – review & editing:** Esete Habtemariam Fenta, Berhan Tassew, Admas Abera, Firmaye Bogale Wolde, Meseret Legesse, Justin Pulford, Siobhan Mor, Mirgissa Kaba.

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
