## [Editor Report · Decision Letter 0]

9 Nov 2023

PONE-D-23-27438Sustaining essential healthcare services in Addis Ababa during COVID-19: A qualitative study.PLOS ONE

Dear Dr. Woldehanna,

Thank you for submitting your manuscript to PLOS ONE. After careful consideration, we feel that it has merit but does not fully meet PLOS ONE’s publication criteria as it currently stands. Therefore, we invite you to submit a revised version of the manuscript that addresses the points raised during the review process.

We look forward to receiving your revised manuscript.

Kind regards,

Admasu Belay Gizaw, MSc

Academic Editor

PLOS ONE

Journal Requirements:

Whilst you may use any professional scientific editing service of your choice, PLOS has partnered with both American Journal Experts (AJE) and Editage to provide discounted services to PLOS authors. Both organizations have experience helping authors meet PLOS guidelines and can provide language editing, translation, manuscript formatting, and figure formatting to ensure your manuscript meets our submission guidelines. To take advantage of our partnership with AJE, visit the AJE website (http://aje.com/go/plos) for a 15% discount off AJE services. To take advantage of our partnership with Editage, visit the Editage website (www.editage.com) and enter referral code PLOSEDIT for a 15% discount off Editage services. If the PLOS editorial team finds any language issues in text that either AJE or Editage has edited, the service provider will re-edit the text for free.

This research was supported by the Global Challenges Research Fund (GCRF) One Health Regional Network for the Horn of Africa (HORN) Project, from UK Research and Innovation (UKRI) and Biotechnology and Biological Sciences Research Council (BBSRC) (project number BB/P027954/1). The funders had no role in study design, data collection and analysis, decision to publish, or preparation of the manuscript.

Additional Editor Comments:

Review results of manuscript entitled ‘sustaining essential healthcare services in Addis Ababa during COVID-19: A qualitative study’.

I am happy to review an interesting manuscript. For the better improvements and consideration, I recommended getting a clear answer for the following issues before proceeding to the next process.

1. What about feasibility of this study in relation to the current status of COVID-19 and re-initiation of essential healthcare services.

2. What is the novelty of this study?

a) What is already known about the problem?

b) What is not known about the problem?

c) What is the possible implication/s for practice?

3. What does it mean by sustaining essential healthcare services?

4. What does it mean by health system’s response to maintaining essential healthcare services and sustaining essential healthcare services?

5. What about validity and reliability of the data collection tools?

6. The analysis method is not clear and strong!

7. What are the main findings & conclusion from this study? Result or hypothesis?

8. The study uses the small number of participants; why?

9. What is the relationship between the objective and the result section?

10. The conclusion section is difficult to understand and very crude and vague for understanding.

11. Which citation style the authors used? Clarifying and revision may be helpful.

12. Considering English grammar and language revision is also very important.

*Author need to clarify the above points via point by point response just before considering the next review

---

## [Author Response · Author response to Decision Letter 0]

20 Dec 2023

12/20/2023

PLOS ONE

Dear Editor, 

We thank you for the opportunity to revise and resubmit this manuscript. We also appreciate constructive comments and recommendations from the reviewer. Below are our responses to the comments from the reviewer and a description of the changes we have made (texts in bullet points and italic font). 

Reviewer comment 

1. What about feasibility of this study in relation to the current status of COVID-19 and re-initiation of essential healthcare services.

• The feasibility of this study is reflected in the fact that though COVID-19 is no longer a pandemic, we can use the findings of this study as input for inferring as a learning point to similar public health situations. Additionally, the adaptation strategies identified in this study can be used for emergency preparedness programs and decision-making. This has been described in the conclusion section of the manuscript (lines 494-496). 

2. What is the novelty of this study?

a) What is already known about the problem?

b) What is not known about the problem?

c) What is the possible implication/s for practice?

• This study is novel because it has studied the less explored effect of COVID-19 on essential health service provision in health facilities of urban settings serving a large number of populations. The context of the study is emphasized in this study to improve the applicability of findings to similar public health situations in relation to essential healthcare services. Thus, it highlights the existing knowledge base on the effect of the pandemic on health systems as a whole and narrows in the specifics of the selected setting and phenomena for future contextualization. This is mainly highlighted in the final paragraph of the background section of the manuscript (line 83-93). 

3. What does it mean by sustaining essential healthcare services? 

• Sustaining essential healthcare services in this study is used to show the provision of uninterrupted essential healthcare services to communities. This is defined in the third paragraph of the background section on line 72-73. Note: we have used the term sustaining and maintaining essential healthcare services interchangeably in this manuscript. To avoid confusion, we have now used the word maintaining throughout. 

4. What does it mean by health system’s response to maintaining essential healthcare services and sustaining essential healthcare services?

• The concept of the health system’s response to maintaining essential healthcare services indicates actions taken by the different levels/parts of the health sector to ensure the continued provision of health services, where the statement “maintaining essential healthcare services” holds the same meaning. However, as mentioned in the response to the above question, we have used the terms sustaining and maintaining interchangeably, which can be confusing. Hence, to avoid any confusion and to align with documents of the same concept, we have now used uniform wording, i.e., maintaining essential healthcare services. This is shown in the description of the aim of the study in the last sentence of the Background section of the manuscript.

5. What about validity and reliability of the data collection tools?

• We pilot-tested interview guides and made the necessary refinements before data collection to ensure the validity and reliability of data collection tool. We’ve included this under Data collection and Procedure (line 125-129). 

6. The analysis method is not clear and strong!

• We have edited the analysis method for clarity (line 139-151). 

7. What are the main findings & conclusion from this study? Result or hypothesis?

• The study highlights the effect of COVID-19 on the health system and the health sector's action to ensure the continued provision of essential health services during the pandemic's initial phase. The main findings of this study are summarized in the first paragraph of the discussion. (line 430-442). 

8. The study uses small number of participants; why?

• We determined the sample size based on the saturation of information. We have clarified this in (line 117-118). 

9. What is the relationship between the objective and the result section?

• This study was conducted with the aim of exploring the actions taken by the health sector to ensure sustained provision of essential healthcare services during the COVID-19 pandemic. With this study, we were able to highlight the impact of COVID-19 on service utilization and provision. The result also highlighted key adaptation strategies implemented by the health sector to sustain the provision of essential healthcare services and the challenges encountered. These key findings have been reported in the result section and were discussed accordingly. 

10. The conclusion section is difficult to understand and very crude and vague for understanding. 

• We have edited the conclusion section for clarity (line 495-505). 

11. Which citation style the authors used? Clarifying and revision may be helpful. 

• We have used the Vancouver style for citation. Revision to the reference section has been made accordingly. 

12. Considering English grammar and language revision is also very important.

• We have made edits to the manuscript for English grammar and language. All authors reviewed and edited the manuscript.

---

## [Decision Letter · Decision Letter 1]

12 Apr 2024

PONE-D-23-27438R1Maintaining essential healthcare services in Addis Ababa during COVID-19: A qualitative study.PLOS ONE

Dear Dr. Woldehanna,

Thank you for submitting your manuscript to PLOS ONE. After careful consideration, we feel that it has merit but does not fully meet PLOS ONE’s publication criteria as it currently stands. Therefore, we invite you to submit a revised version of the manuscript that addresses the points raised during the review process.

We look forward to receiving your revised manuscript.

Kind regards,

Faten Amer, PhD in Health Sciences

Academic Editor

PLOS ONE

Additional Editor Comments:

The manuscript under review presents a comprehensive qualitative analysis of the impact of COVID-19 on healthcare service utilization and provision in Addis Ababa, Ethiopia. It meticulously examines the challenges encountered by the healthcare system, encompassing reduced service utilization, provider apprehension, and compromised service quality. Despite notable strengths, such as the inclusion of stakeholders from diverse levels of the healthcare system and the provision of rich qualitative data, the study falls short of meeting several criteria outlined by the COREQ (Consolidated Criteria for Reporting Qualitative Research) checklist. Consequently, the authors are encouraged to address these deficiencies to enhance the rigor and transparency of their study.

Domain 1: Research Team and Reflexivity

- Interviewer/Facilitator: Interviews were conducted with participants, but details regarding the credentials, occupation, gender, experience, training, and relationship establishment of the interviewers/facilitators remain undisclosed.

Domain 2: Study Design

- Theoretical Framework: The manuscript lacks explicit reference to a theoretical framework, which is pivotal for contextualizing the study.

- Participant Selection: Purposive sampling was employed, yet details regarding the method of approach, sample size, non-participation, setting of data collection, presence of non-participants, and description of the sample are absent.

Domain 3: Analysis and Findings

- Interview Guide: The development and pilot testing of interview guides are not addressed, which is crucial for ensuring data collection validity.

- Data Analysis: Although themes were derived from the data, information regarding the number of data coders, description of the coding tree, software used for analysis, and participant checking is notably absent.

Reporting

- Quotations Presented: Participant quotations are utilized to illustrate themes/findings, but consistency between data presented and findings, clarity of major and minor themes, and discussion of diverse cases warrant improvement for enhanced clarity and transparency.

Conclusion

Overall, while the manuscript provides valuable insights into the effects of COVID-19 on healthcare in Addis Ababa, Ethiopia, it is essential for the authors to address the aforementioned deficiencies outlined by the COREQ checklist to strengthen the study's methodological rigor and reporting clarity. By doing so, the manuscript would significantly contribute to the existing literature on pandemic response strategies in healthcare settings, thereby

Reviewers' comments:

Reviewer's Responses to Questions

**Comments to the Author**

1. If the authors have adequately addressed your comments raised in a previous round of review and you feel that this manuscript is now acceptable for publication, you may indicate that here to bypass the “Comments to the Author” section, enter your conflict of interest statement in the “Confidential to Editor” section, and submit your "Accept" recommendation.

Reviewer #1: (No Response)

2. Is the manuscript technically sound, and do the data support the conclusions?

Reviewer #1: Yes

3. Has the statistical analysis been performed appropriately and rigorously? 

Reviewer #1: N/A

4. Have the authors made all data underlying the findings in their manuscript fully available?

Reviewer #1: (No Response)

5. Is the manuscript presented in an intelligible fashion and written in standard English?

Reviewer #1: Yes

6. Review Comments to the Author

**Reviewer #1:** I have a few concern and questions on this work:

1. Why are the authors interested to employ only key informant interviews? Why are they interested in interviewing only "highly knowlesgeable" respondents? What does highly knowledgeable even mean? For example let me just use one of your findings to explain my concern. It was found out that discontinuance of service utilization (such as ART follow_up). Were you thinking that the drivers and means of mitigation of such discontinuece is most understood and addressed through the recomendations and ideas of highly knowledgeable health system workers alone? What do you think was the role of discussing with communities or beneficiaries (through indepth interviews and FGDs)? Do not you think missing relevant ideas from such population would have benefited the study? Moreover, do not you think that mitigations and maintainace of essential services are collective and participantory responses whereby community engagement should be manifested in? Hence, I see some limitation on the study design and methods which should be honestly reported in the study.

2. Can you please give a separate section in the methods for "rigor"? Wherein you discuss the main dimensions of trusthworthineass of the study and how you implemented them: crediblity, depedabality, transferrablity, and conformablity

Thank you for the great job

7. PLOS authors have the option to publish the peer review history of their article (what does this mean?). If published, this will include your full peer review and any attached files.

Reviewer #1: No

---

## [Author Response · Author response to Decision Letter 1]

29 May 2024

5/25/2024

PLOS ONE 

Dear Editor, 

We thank you for the opportunity to revise and resubmit this manuscript. We also appreciate constructive comments and recommendations from the reviewer. Below are our responses to the comments from the reviewer and a description of the changes we have made (texts in bullet points and italic font). 

Editor’s comment 

Domain 1: Research Team and Reflexivity

- Interviewer/Facilitator: Interviews were conducted with participants, but details regarding the credentials, occupation, gender, experience, training, and relationship establishment of the interviewers/facilitators remain undisclosed.

- Thank you for your comments. We have included the details of the interviewers in the methods section (line 143-147).

Domain 2: Study Design

- Theoretical Framework: The manuscript lacks explicit reference to a theoretical framework, which is pivotal for contextualizing the study.

- Thank you for the comments. We have used an exploratory study design to guide this study. We have now included the study design in the method section under study design and setting section (line 97-100).

- Participant Selection: Purposive sampling was employed, yet details regarding the method of approach, sample size, non-participation, setting of data collection, presence of non-participants, and description of the sample are absent.

- Thank you for the comments. We have included the method of approach, sample size, non-participation, setting of data collection, presence of non-participants, and the study size in the method section under Sample size and study participants (line 114-126). Furthermore, the sample description is now included in the first part of the result (line 199-202).

Domain 3: Analysis and Findings

- Interview Guide: The development and pilot testing of interview guides are not addressed, which is crucial for ensuring data collection validity.

- The aspects of interview guide development, components of the guide, pilot testing and changes as a result of pilot testing are addressed in the “Data collection tool and procedure” sections (line 129-138)

- Data Analysis: Although themes were derived from the data, information regarding the number of data coders, description of the coding tree, software used for analysis, and participant checking is notably absent.

- Thank you for your comments. We have addressed issues of coding in the methods and materials section under data analysis and we have provided description of the coding tree at the beginning of the results section (line 203-108). The software used for analysis is also indicated in the analysis section (line 165). We have now included details of rigor in the Methods and materials section under rigor (line 168-184).

Reporting

- Quotations Presented: Participant quotations are utilized to illustrate themes/findings, but consistency between data presented and findings, clarity of major and minor themes, and discussion of diverse cases warrant improvement for enhanced clarity and transparency.

- Thank you for the important insight. We have indicated the themes and sub-themes to show the major and minor themes and their relationship, we have also included the sources of the quotations to create more clarity and transparency. 

Reviewers' comments

I have a few concern and questions on this work:

1. Why are the authors interested to employ only key informant interviews? Why are they interested in interviewing only "highly knowlesgeable" respondents? What does highly knowledgeable even mean? For example let me just use one of your findings to explain my concern. It was found out that discontinuance of service utilization (such as ART follow_up). Were you thinking that the drivers and means of mitigation of such discontinuece is most understood and addressed through the recomendations and ideas of highly knowledgeable health system workers alone? What do you think was the role of discussing with communities or beneficiaries (through indepth interviews and FGDs)? Do not you think missing relevant ideas from such population would have benefited the study? Moreover, do not you think that mitigations and maintainace of essential services are collective and participatory responses whereby community engagement should be manifested in? Hence, I see some limitation on the study design and methods which should be honestly reported in the study.

- Thank you for the valuable comment and insight. We accept that this study would have been more comprehensive if the community’s perspective had been included. Thus, we have put this as a limitation of this study (line 539-540). 

- With the word “knowledgeable”, we were trying to indicate that the participants possibly had comprehensive information about the effect of COVID-19 and the mitigation strategies and/ or the health system response to sustain the provision of essential healthcare services in their respective departments. Thus, we have removed the term knowledgeable to avoid ambiguity and replaced it with the meaning it portrays (line 114-118).

2. Can you please give a separate section in the methods for "rigor"? Wherein you discuss the main dimensions of trusthworthineass of the study and how you implemented them: crediblity, depedabality, transferrablity, and conformablity

Thank you for the great job

 - We have now created a separate rigor section in the methods and materials and included details about activities conducted to ensure trustworthiness (line 168-184)

---

## [Decision Letter · Decision Letter 2]

25 Jul 2024

Maintaining essential healthcare services in Addis Ababa during COVID-19: A qualitative study.

PONE-D-23-27438R2

Dear Dr. Woldehanna,

We’re pleased to inform you that your manuscript has been judged scientifically suitable for publication and will be formally accepted for publication once it meets all outstanding technical requirements.

Kind regards,

Faten Amer, PhD in Health Sciences

Academic Editor

PLOS ONE

Additional Editor Comments (optional):

Reviewers' comments:

Reviewer's Responses to Questions

**Comments to the Author**

1. If the authors have adequately addressed your comments raised in a previous round of review and you feel that this manuscript is now acceptable for publication, you may indicate that here to bypass the “Comments to the Author” section, enter your conflict of interest statement in the “Confidential to Editor” section, and submit your "Accept" recommendation.

Reviewer #1: All comments have been addressed

2. Is the manuscript technically sound, and do the data support the conclusions?

Reviewer #1: Yes

3. Has the statistical analysis been performed appropriately and rigorously? 

Reviewer #1: N/A

4. Have the authors made all data underlying the findings in their manuscript fully available?

Reviewer #1: (No Response)

5. Is the manuscript presented in an intelligible fashion and written in standard English?

Reviewer #1: (No Response)

6. Review Comments to the Author

Reviewer #1: (No Response)

7. PLOS authors have the option to publish the peer review history of their article (what does this mean?). If published, this will include your full peer review and any attached files.

Reviewer #1: No

---

## [Editor Report · Acceptance letter]

5 Aug 2024

PONE-D-23-27438R2 

PLOS ONE

Dear Dr. Woldehanna, 

I'm pleased to inform you that your manuscript has been deemed suitable for publication in PLOS ONE. Congratulations! Your manuscript is now being handed over to our production team.

Kind regards, 

on behalf of

Dr. Faten Amer 

Academic Editor

PLOS ONE